# Grid-Based Evolutionary Algorithm for Multi-Objective Molecule Generation Enhanced by Reinforcement Learning

## Abstract

Fragment-based drug discovery (FBDD) is limited by the need to construct and maintain static fragment libraries. To overcome these challenges, we propose a novel evolutionary framework. Our method starts with sample molecules that are fragmented and fed into a policy-decoupled architecture. This architecture utilizes reinforcement-learning-guided crossover and mutation operators to recombine and modify fragments. This approach expands the latent fragment space without relying on predefined libraries. By employing a grid-based fragment-masked crossover, our method enables combinatorial explorations and extends beyond conventional fragmentation patterns. In comparative experiments, our method outperforms recent state-of-the-art methods on most PMO benchmarks and target-protein docking tasks. Additionally, it achieves a low average synthetic accessibility (SA) score and maintains a structural novelty rate above 90%.

## 1 Introduction

Computer-aided molecular generation has emerged as a crucial pillar of contemporary drug discovery. It capitalizes on advancements in deep learning and evolutionary computation to navigate extensive chemical spaces, offering greater efficiency than traditional high-throughput screening and medicinal chemistry methods (Xue et al., 2025). Among these strategies, fragment-based drug design (FBDD) stands out by integrating low-molecular-weight fragments into lead compounds that exhibit high affinity, potency, and favorable drug-like properties (Maziarz et al., 2021). This methodology leverages the synthetic tractability and diverse binding modes of small fragments. It applies structure-guided optimization to achieve high precision. The balance between hit rate, chemical diversity, and synthetic feasibility makes FBDD a widely adopted approach (Geng et al., 2023).

Over the last decade, numerous implementations of FBDD have been proposed (Xie et al., 2021; Jensen, 2019; Tripp & Hernández-Lobato, 2023). The processes of these methods include: (1) constructing comprehensive fragment vocabularies via extraction techniques (Yang et al., 2021; Xie et al., 2021; Maziarz et al., 2021), (2) maintaining and expanding these vocabularies through fragment modification (Jensen, 2019; Tripp & Hernández-Lobato, 2023; Lee et al., 2024b;a), and (3) employing deep neural architectures to explore combinatorial assembly (Jin et al., 2020b; Maziarz et al., 2021; Kong et al., 2022; Geng et al., 2023; Yang et al., 2021; Lee et al., 2024b;a).

Nevertheless, they face several critical limitations: (i) reliance on predefined fragment libraries inherently constrains chemical exploration; (ii) construction and maintenance of these libraries incur significant inefficiencies (Wang et al., 2022); and (iii) integration of fragments into deep generative models often yields a black-box generation process, reduces the interpretability and complicates multi-property balancing (Angelo et al., 2023).

To address these limitations, we propose a Reinforcement Learning-Driven Grid-based Fragment-Masked Multi-objective Evolutionary Algorithm (RL-GFM). RL-GFM obviates the need to predefine and maintain the fragment libraries by dynamically recombining and modifying molecules through variously designed crossover and mutation operators.

The main contributions of this paper can be summarized as follows:

- **RL-based crossover and mutation**: We employ a policy-decoupled architecture, wherein two actor–critic reinforcement learning (RL) agents specialize in operator coordination. By integrating multi-objective dominance information into policy backpropagation, these agents learn to select mutation rules for the mutation operator and pair elite parents for the crossover operator. This design avoids reliance on random-walk-like exploration, thereby mitigating the risk of unstable performance.

- **Grid-based fragment-masked crossover**: To transcend conventional fragmentation patterns, we partition molecules into spatial grid regions and enforce masked bond-cleavage rules. Reference points guide parent pairing to maximize inter-grid diversity and intra-grid convergence. Non-scaffold single bonds are randomly cleaved and recombined according to validated dissociation/recombination protocols. Constraining randomness to pharmacophore-compatible regions enables the algorithm to escape local optima and generate synthetically tractable, structurally novel candidates (achieving $\geq 90\%$ novelty).

- **Interpretability and Multi-objective optimization capabilities**: The core generation process of our method is grounded in a multi-objective evolutionary algorithm (MOEA), which inherently incorporates multi-objective optimization capabilities. To avoid sacrificing the primary objective, we employ distinct functional strategies tailored to different tasks. Moreover, by enabling explicit operations on SMILES strings (e.g., bond cleavage and recombination rules, Krenn et al. (2022)), the algorithm renders the generation process interpretable.

The remainder of this article is organized as follows. Section 2 reviews related work, Section 3 details the design and implementation of our RL-GFM framework, Section 4 evaluates our method on PMO and target-protein docking tasks, and Section 5 concludes with future directions.

## 2 RELATED WORK

FBDD has emerged as a prominent strategy in molecular generation. Early approaches primarily relied on constructing fragment libraries through extraction methods (Yang et al., 2021; Xie et al., 2021; Maziarz et al., 2021). However, the use of static libraries inherently restricts the chemical space that can be explored. Even recent FBDD-related methods Tan et al. (2022); Khemchandani et al. (2020) that integrate reinforcement learning or graph models still fail to address these three core limitations: they either rely on predefined fragment/graph templates (limitation (i)), lack interpretable assembly rules (limitation (iii)), or neglect efficient multi-property balancing (Mercado et al., 2021). This underscores the need for a dynamic, interpretable framework like our RL-GFM.

To address this, more recent methods (Tripp & Hernández-Lobato, 2023; Lee et al., 2024b;a) employ genetic algorithms with crossover and mutation operators to dynamically evolve fragment combinations. These approaches typically optimize either a single molecular property or combine multiple objectives into a single scalar function, often neglecting the inherent trade-offs among objectives (Xie et al., 2021). In contrast, Verhellen et al. (Verhellen, 2022) employed the Non-dominated Sorting Genetic Algorithm II (NSGA-II) algorithm to perform multi-objective molecular optimization. This approach decomposes parent molecules (SMILES strings) into fragments, generates offspring molecules through recombining and modifing these fragments. The generated molecules are selected by non-dominated sorting and crowding distance computations. Unlike methods that depend on predefined fragment vocabularies, this strategy leverages evolutionary principles to navigate the chemical space. Despite its strengths in multi-objective optimization and interpretability, this method remains vulnerable to issues such as stochasticity and premature convergence to local optima, which can hinder its ability to transcend existing fragment and assembly models.

To overcome these limitations, we propose the RL-GFM framework, which incorporates RL agents to guide search, a grid-based fragment-masked crossover mechanism to transcend existing fragments, and a core process governed by a MOEA for balancing multiple objectives. By integrating these components, RL-GFM delivers more consistent and robust performance in complex molecular discovery tasks.

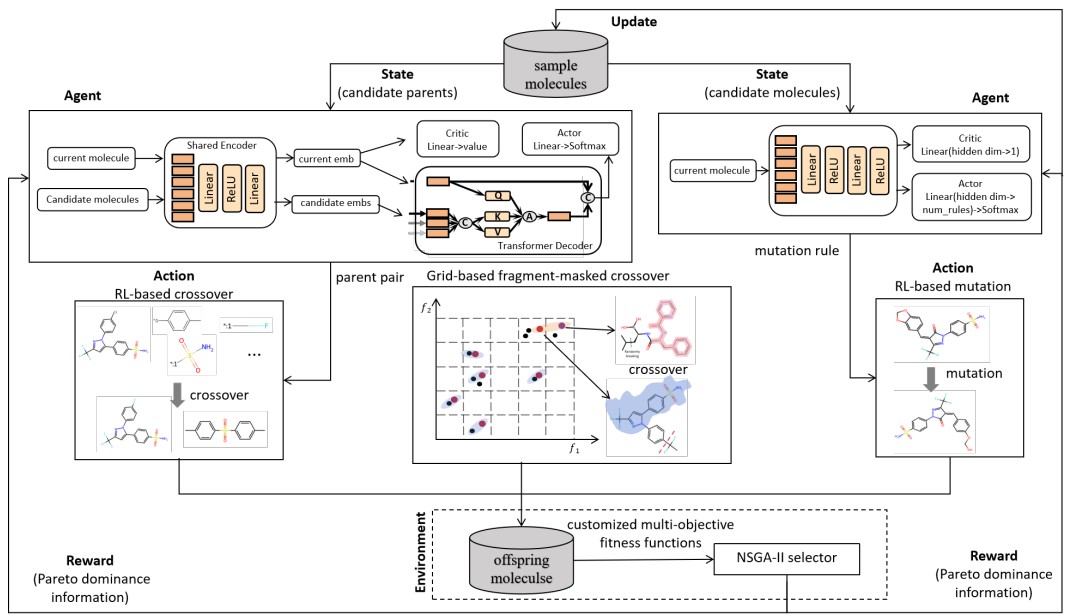

Figure 1: Workflow of the RL-GFM Algorithm. The population is initialized from ZINC250K; molecules are generated via three kinds of operators: grid-based fragment-masked crossover, RL-based crossover, and RL-based mutation. Finally, all molecules are filtered down to $N$ molecules for the next generation by the NSGA-II selector paired with customized multi-objective functions..

## 3 METHODS

The proposed RL-GFM framework integrates adaptive policy control with fragment-guided structural exploration. It comprises three core components: (1) a grid-based fragment-masked crossover operator, which facilitates diversity-driven recombination; (2) RL-driven crossover and mutation operators, designed to enable targeted exploration; and (3) a Pareto-optimal solution selector paired with customized multi-objective functions to identify solutions with balanced objectives. The framework's overall workflow is depicted in Figure 1.

### 3.1 GRID-BASED FRAGMENT-MASKED CROSSOVER

Grid-based strategies are commonly employed as a solution selection mechanism in multi-objective optimization, where partitioning the objective space into multiple grid cells serves to enhance solution diversity (Yang et al., 2013; Xu et al., 2023). Within RL-GFM, we adapt this grid-based strategy for parent pairing by defining two types of specialized reference points—grid ideal points and global ideal points—that guide the parent pairing process. Leveraging these paired parent molecules, we then implement a fragment-masked crossover mechanism, which further enhances chemical structural diversity and strengthens the algorithm's capacity to explore the chemical space.

**The grid-based parent pairing method** organizes candidate solutions in the objective space to balance diversity and convergence. First, for each objective dimension $f_k$ $(k = 1, 2, \ldots, M)$, the maximum observed value $f_k^{\max}$ is determined and the interval $[0, f_k^{\max}]$ is divided into five equal sub-intervals. Their Cartesian product then defines $5^M$ hypercubes (grid cells). Each solution $x$ is assigned to a cell via $\mathcal{G}(x) = \bigotimes_{k=1}^{M} \left\lfloor \frac{f_k(x)}{f_k^{\max}/5} \right\rfloor$.

In cases where a grid cell contains only a single solution, that solution is reassigned to its nearest non-empty neighboring grid. For each non-empty grid $G_i$, we compute the Euclidean distance $d(x) = \sqrt{\sum_{k=1}^{M} f_k(x)^2}$ for all $x \in G_i$, and designate the solution with the maximum $d(x)$ as the grid's ideal point $\mathcal{I}_i$. The pairing mechanism is implemented hierarchically: locally, each solution within $G_i$ is paired with its grid's ideal point to preserve diversity; globally, all grid ideal points

are paired with a global ideal point $\mathcal{I}_g$ (selected from the grid with the highest index) to promote convergence. This dual-level strategy ensures both intra-grid diversity and inter-grid convergence, thereby allocating resources in proportion to grid density (see Figure 2a).

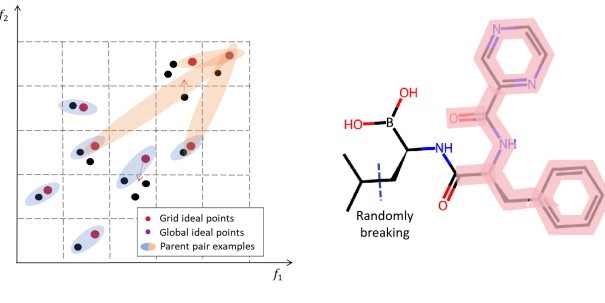

(a) Grid-based parent pairs      (b) Masked molecule

Figure 2: Grid-based fragment-masked crossover. (a) Each solution within a grid is paired with its respective grid's ideal point; all grid ideal points are then paired with the global ideal point. (b) Bonds internal to the scaffold or those connecting the scaffold and side chains are assigned a mask value of 0 (highlighted in pink), while bonds within the side chains are assigned a mask value of 1 (not highlighted), where a random cut can be made.

**The fragment-masked crossover mechanism** is introduced to balance structural preservation and chemical diversity, building on the grid-based parent pair approach. Inspired by Li et al. (2024)(Li et al., 2024), this process begins by dividing each molecule into two functional components, as illustrated in Figure 2b. The protected scaffold (highlighted in pink) comprises the core structural motifs essential for molecular stability and bioactivity, while the side chains (not highlighted) represent modifiable peripheral substituents. Bonds are categorized according to their roles: those internal to the scaffold or connecting the scaffold and side chains are assigned a mask of 0, whereas bonds within the side chains receive a mask of 1. A randomized cleavage is then applied to the regions with mask 1, splitting each molecule into fragments. Fragments from grid-based paired parent molecules are recombined, enabling exploration of under-explored regions of chemical space. By protecting the core scaffold and exposing peripheral substituents for randomized bond cleavage, this mechanism maintains synthetic accessibility and bioactivity while enabling controlled chemical diversity.

### 3.2 RL-BASED CROSSOVER & MUTATION

In our framework, two specialized reinforcement learning (RL) policies are introduced to guide molecular evolution. The goal is to train an actor (the policy, denoted as $\pi$) to select optimal actions (crossover or mutation), and a critic (the value function, denoted as $V$) to evaluate the quality of the current molecular state. Both policies are optimized using an actor-critic method, where their parameters, $\theta_a$ for the actor and $\theta_c$ for the critic, are updated based on a reward signal $r_i$ derived from the quality of the generated offspring.

**RL-based Crossover Policy:** The crossover actor $\pi^\times(a_{i,j} \mid s_i; \theta_a)$ is a transformer-decoder with cross-attention designed to select an appropriate parent molecule $a_{i,j}$ to cross with the current molecule $s_i$. First, a shared two-layer MLP encodes the Morgan fingerprint $(\mathrm{FP}(\cdot))$ of the current molecule $(s_i)$ and each candidate parent $(a_{i,j})$ into latent vectors. The decoder then computes pairwise compatibility scores via cross-attention, and a softmax function yields the selection probabilities. The crossover critic is a single linear layer applied to the shared encoder output. It is formulated as $V^\times(s_i; \theta_c) = w_c^\top \mathrm{Enc}(s_i)$, where $\mathrm{Enc}(s_i)$ is the latent vector for state $s_i$ produced by the encoder.

**RL-based Mutation Policy:** The mutation actor $\pi^\mu(a_{i,j} \mid s_i; \theta_a)$ is a two-layer ReLU-MLP that outputs a probability distribution over a set of predefined mutation operations $a_{i,j}j = 1^P$. Its hidden state $h_i$ is computed as $h_i = \mathrm{ReLU}(W_2, \mathrm{ReLU}(W_1, \mathrm{FP}(s_i)))$, and the policy is $\pi^\mu(ai, j \mid s_i) = \mathrm{softmax}(W_3, h_i)$. Similarly, its critic $V^\mu(s_i; \theta_c) = w_c'^\top h_i$ is a linear layer operating on $h_i$.

**Loss Formulation:** Both policies are optimized via a unified actor-critic framework. For each policy type $* \in \{\times, \mu\}$, let $\theta_a$ denote the policy-specific actor parameters and $\theta_c$ represent shared

critic parameters. At each training iteration, the actor samples an action $a_{i,j}$ from the candidate set $\{a_{i,j}\}_{j=1}^{Q}$ following the policy $\pi^*(a_{i,j} \mid s_i; \theta_a)$, where $s_i$ is the molecular state and $a_{i,j}$ corresponds to either a crossover partner or mutation operation. The loss function for policy $*$ is:

$$\mathcal{L}^*(\theta_a, \theta_c) = \frac{1}{N} \sum_{i=1}^{N} \left[ -\log \pi^*(a_{i,j} \mid s_i; \theta_a) \cdot A_i^* + (V^*(s_i; \theta_c) - r_i)^2 \right] \tag{1}$$

where $A_i^* = r_i - V^*(s_i; \theta_c)$ is the advantage function. The reward $r_i$ is defined as:

$$r_i = \begin{cases} \mathrm{GMean}(\mathbf{f}_{\mathrm{off},i}), & \text{if } i\text{-th offspring is Pareto-optimal,} \\ -\epsilon, & \text{if } i\text{-th offspring is non-Pareto-optimal,} \\ -\mathrm{GMean}(\mathbf{f}_{\mathrm{off},i}), & \text{if same parent pair is used.} \end{cases} \tag{2}$$

with $\mathrm{GMean}(\mathbf{f}) \triangleq \left( \prod_{k=1}^{n} f_k \right)^{1/n}$ and $\epsilon = 10^{-6}$. GMean is chosen to avoid single-objective dominance. This design synergizes with Pareto optimality by quantifying the "dominance strength" of Pareto solutions, guiding the RL policy to prioritize multi-objective balanced parent pairs. The small value of $\epsilon$ avoids overwhelming the policy update with excessive penalty signals for non-Pareto solutions, while still serving as a weak constraint to guide exploration away from low-quality molecules.

The advantage term $A_i^*$ prioritizes actions yielding higher returns than the critic's baseline estimate, while the mean squared error (MSE) term $(V^*(s_i; \theta_c) - r_i)^2$ trains the critic to improve value predictions.

**Crossover and mutation:** The crossover policy determines parent pair molecules, which are then split into fragments and randomly recombined to form new molecules. For mutation, specific mutation rules are selected by the policy; these rules modify the molecules to facilitate exploration of chemical novelty. Their synergistic operation establishes an evolutionary loop consistent with the process described in (Verhellen, 2022).

### 3.3 MULTI-OBJECTIVE OPTIMIZATION

**NSGA-II selector:** After generating molecules via crossover and mutation, these resulting molecules are filtered using the NSGA-II selector. This selector leverages fast non-dominated sorting and crowding-distance computation to enable efficient traversal of the multi-objective search space; its pseudocode is provided in Appendix 1.4.

**Customized objective functions:** We employ two strategies: *complete attribute separation* and *adaptive weighting scheme*. The former guarantees that molecules exhibiting improved performance in each attribute, albeit with the risk of sacrificing primary objective. By contrast, the adaptive weighting scheme treats the primary objective as an independent target while defining composite targets for secondary objectives, wherein adaptive parameters dynamically regulate their trade-offs.

On the PMO benchmark tasks, the adaptive weighting scheme is formulated as follows:

$$[Obj_1, Obj_2] = \left[ \mathrm{score}, \lambda \times \mathrm{score} + \widehat{SA} \right], \lambda = 1 - \frac{1}{1 + \exp\left( -k \left( x - \mathrm{max\_calls}/2 \right) \right)} \tag{3}$$

where $\mathrm{score}$ denotes the PMO task score; $\lambda$ is computed via a sigmoid decay function; and $k = 0.001$ is an attenuation coefficient. The synthetic accessibility $\widehat{SA}$ score is normalized as $\widehat{SA} = \frac{10-SA}{9}$, as defined in (Lee et al., 2024a).

For docking tasks, we adopt the complete attribute separation approach, defining a tri-objective function as:

$$[Obj_1, Obj_2, Obj_3] = \left[ \widehat{DS}, \mathrm{QED}, \widehat{SA} \right] \tag{4}$$

with $\widehat{DS} = -\frac{\mathrm{clip}(DS, -20, 0)}{20}$ in accordance with (Lee et al., 2024a). This explicit separation prevents any single factor from dominating the selection process.

Table 1: Baseline methods employed for experimental comparison, classified by representation modalities (columns) and optimization algorithms (rows).

| | SMILES | SELFIES | Graph (fragment) |
|---|---|---|---|
| EA | Mol-GA/NSGA-II | STONED/GA+D | Graph-GA |
| BO | | | GP-BO |
| SBM | Genetic-GFN/GEGL | | MARS |
| HC | LSTM-HC | | |
| RL | REINVENT | SELFIES-REINVENT | MORLD/GEAM/RationaleRL/FREED |
| VAE | | | JT-VAE/HierVAE/PS-VAE |
| DM | | | MOOD |
| Transformer | RetMol | | f-RAG |

In the PMO benchmark, we prioritize task-specific optimization above all else, with SA serving as a secondary consideration; accordingly, we employ an adaptive weighting scheme for the evaluation function. By contrast, in the molecular docking task, we regard docking score, SA, and Quantitative Estimate of Drug-likeness (QED) as equally critical, and thus adopt a fully decoupled attribute-separation approach.

## 4 EXPERIMENTS

In this section, we extensively evaluate RL-GFM across multiple molecular optimization tasks designed to reflect diverse challenges in drug discovery.

Table 1 provides a comprehensive overview of the algorithms used in the experimental evaluation. The methodologies include three molecular representation modalities: SMILES strings, SELFIES strings (Krenn et al., 2020), and graph (fragment)-based representations. Additionally, eight distinct optimization strategies are employed: Evolutionary Algorithms (EA), Bayesian Optimization (BO), Score-Based Modeling (SBM), Hill Climbing (HC), Reinforcement Learning (RL), Variational Autoencoders (VAE), Diffusion Models (DM), and Transformer.

For all tasks, molecules are drawn from a standard repository (ZINC250k (Irwin et al., 2012)). All molecules are filtered according to the rules in (Verhellen, 2022).

### 4.1 EXPERIMENTS ON PMO BENCHMARK

**Setup**. We evaluate RL-GFM's performance across all 23 PMO benchmark tasks. Following the benchmark's standard protocol, we cap the number of oracle queries at 3,000 (10,000 for baselines according to Lee et al. (2024a)) and assess optimization efficacy using the area under the curve (AUC) of the average property scores for the top-10 molecules as a function of the number of oracle calls. Beyond optimization, we assess three critical drug-discovery criteria—diversity, novelty, and synthesizability (SA)—of the generated compounds. Diversity is quantified using the Therapeutics Data Commons (TDC) library (Huang et al., 2021). Novelty is defined as the proportion of generated molecules whose maximum Tanimoto similarity to any molecule in ZINC250k is below 0.4. SA is measured by the standard Synthetic Accessibility Score, where lower scores indicate easier synthesis. All the experiment configuration followed (Lee et al., 2024a) unless otherwise noted.

**Baseline.** We selected nine algorithms from Table 1 as our baseline set: the six top-performing methods reported by the PMO benchmark—REINVENT (Olivecrona et al., 2017), Graph-GA (Jensen, 2019), SELFIES-REINVENT, GP-BO (Tripp et al., 2021), STONED (Nigam et al., 2021), and LSTM-HC (Brown et al., 2019)—together with two recent state-of-the-art approaches, f-RAG (Lee et al., 2024a) and Genetic-GFN (Kim et al., 2024), and the highly effective GA-based Mol-GA (Tripp & Hernández-Lobato, 2023).

**Results.** The main experimental results are summarized in Figure 3. For visual clarity, the figure compares RL-GFM against the five top-performing methods from the original PMO benchmark. Our method demonstrates highly competitive performance across the 23 benchmark tasks. The complete statistical results for the comparison against all nine state-of-the-art baselines are detailed in the Appendix 1.5. Our proposed algorithm achieves superior performance in the majority of cases, attaining the highest cumulative score of 18.007 across all benchmarks, representing a

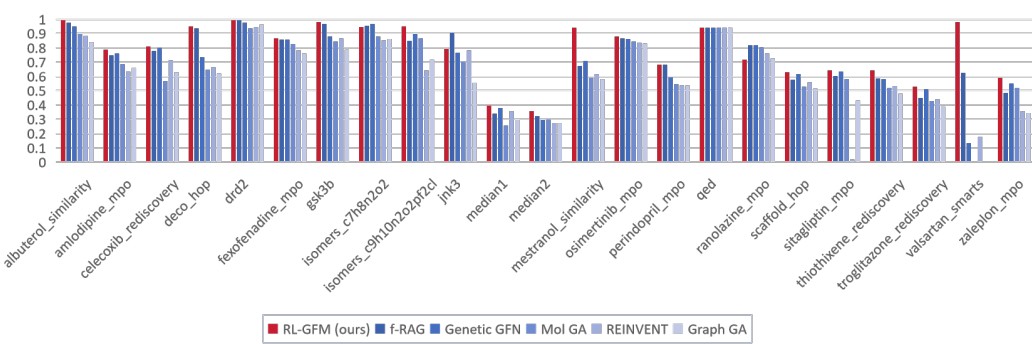

Figure 3: Statistical analysis of top-10 PMO AUC scores.

6.4% improvement over the second-best performing method (f-RAG) and 47.3% over the weakest baseline (LSTM-HC). Notably, RL-GFM exhibits particular dominance in similarity-based tasks such as albuterol_similarity (0.995 vs 0.977 for f-RAG) and mestranol_similarity (0.939 vs 0.708 for Genetic-GFN), suggesting enhanced capability in structural analog generation. The method also excels in complex multi-property optimization (MPO) challenges, including amlodipine_mpo, fexofenadine_mpo and zaleplon_mpo, where it outperforms all baselines by significant margins. A remarkable achievement is observed in the valsartan_smarts task, where RL-GFM attains near-perfect performance (0.979) compared to the zero performance of most algorithms, such as Mol-GA (0.000), Graph-GA (0.000), and GP-BO (0.000), demonstrating exceptional SMARTS pattern matching capabilities.

The comprehensive evaluation of molecular design efficiency across 23 PMO benchmarks reveals the trade-offs between chemical diversity, structural novelty, and SA. As shown in Table 2, RL-GFM achieves the best balance among these metrics, with the highest average diversity (0.573) and novelty (0.912), while maintaining a competitive SA score of 2.332. This tripartite optimization underscores RL-GFM's ability to explore uncharted chemical space without sacrificing practical synthesizability, a core challenge in de novo drug design.

Table 2: Average top-100 diversity, novelty, and SA scores across 23 tasks (best in bold).

| Metric | RL-GFM (ours) | Genetic GFN | Mol GA | REINVENT |
|---|---|---|---|---|
| Average diversity ↑ | **0.573** | 0.443 | 0.491 | 0.468 |
| Average novelty ↑ | **0.912** | 0.724 | 0.845 | 0.540 |
| Average SA score ↓ | **2.332** | 3.770 | 4.605 | 3.207 |

## 4.2 EXPERIMENTS ON DOCKING SCORE

**Setup.** We evaluated RL-GFM on a battery of multi-objective molecular design tasks that seek to maximize binding affinity against a given protein target while simultaneously preserving favorable QED and SA. Binding affinities were approximated by docking scores (in kcal/mol) computed with QuickVina 2 (Alhossary et al., 2015) against five clinically relevant targets—PARP1, FA7, 5HT1B, BRAF, and JAK2—where more negative scores denote stronger predicted interactions. To ensure practical relevance, we quantified QED and SA using the methods described in (Bickerton et al., 2012) and (Ertl & Schuffenhauer, 2009). Consistent with prior work (Lee et al., 2024a), we focused on the top 5% of novel candidates by docking score, with maximum Tanimoto similarity to any molecule in ZINC250k below 0.40, docking score below the median of known actives, QED exceeding 0.50, and SA under 5.0. In all cases, docking scores are reported as mean ± standard deviation over three independent runs, with smaller (more negative) values indicating tighter predicted binding. The Novel hit ratio (%) quantifies the proportion of unique and novel hits within the generated molecular corpus.

**Baselines.** To benchmark performance, we compared RL-GFM against sixteen contemporary generative methods spanning variational autoencoders (JT-VAE (Jin et al., 2018), HierVAE (Jin

Table 3: Novel top 5% docking score (kcal/mol) results. The results are the means and standard deviations of 3 independent runs. The results for compared methods are taken from Lee et al. (Lee et al. 2024a,b). Lower is better, and the best results are highlighted in bold.

| Method | Target protein | | | | |
|---|---|---|---|---|---|
| | parp1 | fa7 | 5ht1b | braf | jak2 |
| JT-VAE | -9.482 ± 0.132 | -7.683 ± 0.048 | -9.382 ± 0.332 | -9.079 ± 0.069 | -8.885 ± 0.026 |
| REINVENT | -8.702 ± 0.523 | -7.205 ± 0.264 | -8.770 ± 0.316 | -8.392 ± 0.400 | -8.165 ± 0.277 |
| Graph GA | -10.949 ± 0.532 | -7.365 ± 0.326 | -10.422 ± 0.670 | -10.789 ± 0.341 | -10.167 ± 0.576 |
| MORLD | -7.532 ± 0.260 | -6.263 ± 0.165 | -7.869 ± 0.650 | -8.040 ± 0.337 | -7.816 ± 0.133 |
| HierVAE | -9.487 ± 0.278 | -6.812 ± 0.274 | -8.081 ± 0.252 | -8.978 ± 0.525 | -8.285 ± 0.370 |
| GA+D | -8.365 ± 0.201 | -6.539 ± 0.297 | -8.567 ± 0.177 | -9.371 ± 0.728 | -8.610 ± 0.104 |
| MARS | -9.716 ± 0.082 | -7.839 ± 0.018 | -9.804 ± 0.073 | -9.569 ± 0.078 | -9.150 ± 0.114 |
| GEGL | -9.329 ± 0.170 | -7.470 ± 0.013 | -9.086 ± 0.067 | -9.073 ± 0.047 | -8.601 ± 0.038 |
| RationaleRL | -10.663 ± 0.086 | -8.129 ± 0.048 | -9.005 ± 0.155 | *No hit found* | -9.398 ± 0.076 |
| FREED | -10.579 ± 0.104 | -8.378 ± 0.044 | -10.714 ± 0.183 | -10.561 ± 0.080 | -9.735 ± 0.022 |
| PS-VAE | -9.978 ± 0.091 | -8.028 ± 0.050 | -9.887 ± 0.115 | -9.637 ± 0.049 | -9.464 ± 0.129 |
| MOOD | -10.865 ± 0.113 | -8.160 ± 0.071 | -11.145 ± 0.042 | -11.063 ± 0.034 | -10.147 ± 0.060 |
| RetMol | -8.590 ± 0.475 | -5.448 ± 0.688 | -6.980 ± 0.740 | -8.811 ± 0.574 | -7.133 ± 0.242 |
| Genetic GFN | -9.227 ± 0.644 | -7.288 ± 0.433 | -8.973 ± 0.804 | -8.719 ± 0.190 | -8.539 ± 0.592 |
| GEAM | -12.891 ± 0.158 | -9.890 ± 0.116 | -12.374 ± 0.036 | -12.342 ± 0.095 | -11.816 ± 0.067 |
| f-RAG | -12.945 ± 0.053 | -9.899 ± 0.205 | -12.670 ± 0.144 | -12.390 ± 0.046 | -11.842 ± 0.316 |
| RL-GFM (ours) | -12.960±0.01 | **-10.310±0.03** | **-13.343±0.2** | **-12.671±0.05** | **-12.090±0.07** |

Table 4: Novel hit ratio (%) results. The results are the means and the standard deviations of 3 runs. The results for the baselines are taken from Lee et al. (Lee et al. 2024b). The best results are highlighted in bold.

| Method | Target protein | | | | |
|---|---|---|---|---|---|
| | parp1 | fa7 | 5ht1b | braf | jak2 |
| REINVENT | 0.480 ± 0.344 | 0.213 ± 0.081 | 2.453 ± 0.561 | 0.127 ± 0.088 | 0.613 ± 0.167 |
| Graph GA | 4.811 ± 1.661 | 0.422 ± 0.193 | 7.011 ± 2.732 | 3.767 ± 1.498 | 5.311 ± 1.667 |
| MORLD | 0.047 ± 0.050 | 0.007 ± 0.013 | 0.880 ± 0.735 | 0.047 ± 0.040 | 0.227 ± 0.118 |
| HierVAE | 0.553 ± 0.214 | 0.007 ± 0.013 | 0.507 ± 0.278 | 0.207 ± 0.220 | 0.227 ± 0.127 |
| RationaleRL | 4.267 ± 0.450 | 0.900 ± 0.098 | 2.967 ± 0.307 | 0.000 ± 0.000 | 2.967 ± 0.196 |
| FREED | 4.627 ± 0.727 | 1.332 ± 0.113 | 16.767 ± 0.897 | 2.940 ± 0.359 | 5.800 ± 0.295 |
| PS-VAE | 1.644 ± 0.389 | 0.478 ± 0.140 | 12.622 ± 1.437 | 0.367 ± 0.047 | 4.178 ± 0.933 |
| MOOD | 7.017 ± 0.428 | 0.733 ± 0.141 | 18.673 ± 0.423 | 5.240 ± 0.285 | 9.200 ± 0.524 |
| GEAM | 40.567 ± 0.825 | 20.711 ± 1.873 | 38.489 ± 0.350 | 27.900 ± 1.822 | 42.950 ± 1.117 |
| RL-GFM (ours) | **43.250 ± 1.041** | **35.570 ± 1.547** | **70.070 ± 5.051** | **42.667 ± 2.245** | 42.557 ± 3.482 |

et al., 2020a), PS-VAE (Kong et al., 2022)), reinforcement-learning and policy-gradient approaches (REINVENT (Olivecrona et al., 2017), RationaleRL (Jin et al., 2020b), GEGL (Ahn et al., 2020), FREED (Yang et al., 2021), MORLD (Jeon & Kim, 2020), MARS (Xie et al., 2021)), genetic-based optimizers (Graph GA (Jensen, 2019), GA + D (Nigam et al., 2020)), as well as more recent methods (RetMol (Wang et al., 2022), MOOD (Lee et al., 2023), Genetic GFN (Kim et al., 2024), GEAM (Lee et al., 2024b), f-RAG (Lee et al., 2024a)).

**Results.** From the results represented in Table 3, our RL-GFM method achieves the most favorable mean docking scores on all five targets, consistently surpassing the previous best-performing baselines (f-RAG and GEAM). Against PARP1, RL-GFM reaches $-12.96 \pm 0.01$ kcal/mol, marginally improving on f-RAG's $-12.945 \pm 0.053$ kcal/mol; the improvement is more pronounced on FA7 ($-10.31 \pm 0.03$ vs. $-9.899 \pm 0.205$ kcal/mol), 5HT1B ($-13.34 \pm 0.20$ vs. $-12.670 \pm 0.144$ kcal/mol), BRAF ($-12.67 \pm 0.05$ vs. $-12.390 \pm 0.046$ kcal/mol), and JAK2 ($-12.09 \pm 0.07$ vs. $-11.842 \pm 0.316$ kcal/mol). These gains—ranging from 0.015 kcal/mol on PARP1 to 0.67 kcal/mol on 5HT1B—demonstrate that RL-GFM systematically identifies higher-affinity chemotypes than the compared baselines.

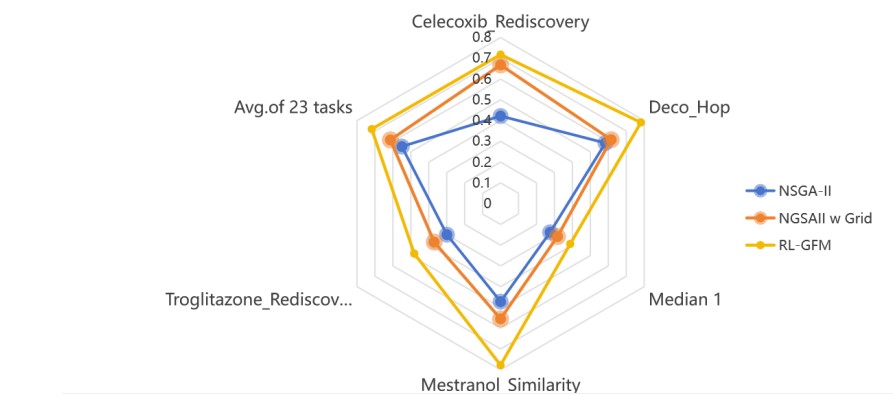

Figure 4: Ablation Study Results. Performance Comparison of NSGA-II Baseline, NSGA-II with Grid-Based Fragment-Masked Crossover (NSGA-II w Grid), and the Full RL-GFM Algorithm on several PMO tasks

Furthermore, we evaluated the model's ability to discover novel chemical matter, measured by the novel hit ratio (%) presented in Table 4. RL-GFM demonstrates exceptional performance, achieving the highest hit ratio on four of the five targets. The improvements over the next-best baseline, GEAM, are particularly striking for 5HT1B, where RL-GFM achieves a $70.070 \pm 5.05$ ratio compared to GEAM's $38.489 \pm 0.350$, and for FA7 ($35.570 \pm 1.547$ vs. $20.711 \pm 1.873$). For the JAK2 target, RL-GFM ($42.557 \pm 3.482$) performs on par with the leading baseline ($42.950 \pm 1.117$). This confirms that our method not only refines molecules towards high-scoring regions but is also highly effective at exploring the chemical space to identify new and diverse molecular structures.

In summary, RL-GFM not only matches or exceeds the state-of-the-art on small-molecule optimization benchmarks but also delivers the strongest predicted binding affinities in structure-based docking tasks, with excellent reproducibility across independent runs.

### 4.3 ABLATION STUDY

To validate the contributions of RL-GFM's core components, we conducted ablation experiments within 2000 function calls across diverse PMO tasks with three experimental groups: NSGA-II (baseline) (Verhellen, 2022), NSGA-II w Grid (NSGA-II integrated with grid-based fragment-masked crossover), and the full RL-GFM.

Results in Figure 4 indicate that NSGA-II w Grid outperforms the NSGA-II baseline across all experimental tasks, including the average score of 23 PMO tasks, confirming that the grid-based crossover effectively enhances chemical space exploration efficiency. Further, the full RL-GFM achieves an additional performance improvement compared to NSGA-II w Grid. This demonstrates that RL-driven operators enable targeted exploration of high-potential molecular regions. Overall, the ablation study verifies the necessity of each component.

### 5 CONCLUSION

In this study, we introduce RL-GFM, a novel framework that integrates innovative crossover and mutation operators within an NSGA-II-based optimization architecture to guide molecular design—thereby mitigating key limitations of fragment-based drug discovery (FBDD), including reliance on predefined and curated libraries, inadequate interpretability, and suboptimal multi-objective balancing. Notably, the RL-driven coordination of crossover and mutation operators enhances adaptive decision-making in navigating trade-offs between objectives, while the grid-based fragment-masked crossover augments both the efficiency and breadth of chemical space exploration. Future work will focus on reconciling RL-based guidance with the inherent stochastic exploration properties of evolutionary algorithms to further refine the balance between targeted search precision and structural diversity in complex chemical spaces.

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
