# GRID-BASED EVOLUTIONARY ALGORITHM FOR MULTI-OBJECTIVE MOLECULE GENERATION ENHANCED BY REINFORCEMENT LEARNING

## 1 APPENDIX

### 1.1 EXPERIMENTAL DETAILS

We performed all RL-crossover and RL-mutation training on a single NVIDIA GeForce RTX 3090 GPU (24 GB). The remaining components of our framework (excluding RL) were executed on an Intel® Xeon® Gold 6230 CPU @ 2.10 GHz with 256 GB of RAM. The RL policies utilized the AdamW optimizer Loshchilov & Hutter with a learning rate of $3e - 4$.

The code of RL-GFM is builded upon the codebase provided by Verhellen et al. Verhellen (2022). The code of fragment-masked part follows the code outlined by Li et al. Li et al. (2024), while our experimental scripts are adapted from Lee et al. Lee et al. (2022; 2023) and Gao et al. Gao et al. (2022). The RL policies use the AdamW optimizer Loshchilov & Hutter with a learning rate of $3e - 4$.

### 1.2 HYPERPARAMETER SENSITIVITY ANALYSIS

To validate the robustness of key hyperparameters and provide guidance for practical adoption, we analyze their impacts.

**Grid Partition Settings.** For the grid partition count in the grid-based crossover operator of RL-GFM, we adopt the configuration of $5^M$, where $M$ denotes the number of optimization objectives. This design is primarily supported by the experimental insights reported by Yang et al. (Yang et al., 2013), which indicate that the number of grid partitions exhibits a strong correlation with the number of objectives. Specifically, when the number of objectives is set to 3, the performance discrepancies observed across partition counts ranging from 5 to 20 are negligible—a finding that is consistent with the results of our own experimental validation. Furthermore, a critical drawback of adopting 10-way or higher partitioning strategies must be emphasized: such configurations tend to result in sparse grid structures. This sparsity not only compromises the diversity of parent selection and pairing (a key factor influencing the exploration capability of evolutionary algorithms) but also significantly increases the computational overhead associated with the parent pairing process. To balance optimization performance, population diversity, and computational efficiency, we ultimately determine 5-way partitioning as the optimal choice for the grid-based crossover operator.

**Decay Coefficient $k$.** The sigmoid-type decay coefficient $k = 0.001$ (see Equation 3) acts as a critical regulatory parameter to achieve balanced optimization between the PMO score and SA. Essentially, the value of $k$ directly determines the weight of SA in the evaluation function, presenting a negative correlation between the magnitude of $k$ and the priority of SA in the overall optimization objectives. Specifically, when users demand enhanced SA performance, the $k$ value can be adjusted to a smaller scale (e.g., $k = 0.0005$. This adjustment accelerates the decay rate of the weight factor $\lambda$ in the sigmoid function: as $\lambda$ decays rapidly, its contribution to the PMO score's weight in the evaluation function is effectively reduced, thereby proportionally enhancing the relative contribution of SA to the overall objective function. This mechanism ensures the optimization process prioritizes SA improvement without compromising the basic structural properties reflected by the PMO score. Notably, this tunable design of $k$ endows the proposed evaluation function with significant flexibility. It allows adaptive adjustment of molecular property priorities to match diverse application scenarios oriented tasks. Such flexibility effectively expands the applicability of the evaluation framework in multi-objective molecular optimization.

Table 1: Statistical analysis of top-10 PMO AUC scores: mean ± standard deviation over three independent runs (best in bold).

| Oracle | RL-GFM (ours) | f-RAG | Genetic-GFN | Mol-GA | REINVENT |
|---|---|---|---|---|---|
| albuterol_similarity | **0.995 ± 0.000** | 0.977 ± 0.002 | 0.949 ± 0.010 | 0.896 ± 0.035 | 0.882 ± 0.006 |
| amlodipine_mpo | **0.788 ± 0.003** | 0.749 ± 0.019 | 0.761 ± 0.019 | 0.688 ± 0.039 | 0.635 ± 0.035 |
| celecoxib_rediscovery | **0.811 ± 0.018** | 0.778 ± 0.007 | 0.802 ± 0.029 | 0.567 ± 0.083 | 0.713 ± 0.067 |
| deco_hop | **0.951 ± 0.001** | 0.936 ± 0.011 | 0.733 ± 0.109 | 0.649 ± 0.025 | 0.666 ± 0.044 |
| drd2 | **0.995 ± 0.000** | 0.992 ± 0.000 | 0.974 ± 0.006 | 0.936 ± 0.016 | 0.945 ± 0.007 |
| fexofenadine_mpo | **0.866 ± 0.002** | 0.856 ± 0.016 | 0.856 ± 0.039 | 0.825 ± 0.019 | 0.784 ± 0.006 |
| gsk3b | **0.980 ± 0.001** | 0.969 ± 0.003 | 0.881 ± 0.042 | 0.843 ± 0.039 | 0.865 ± 0.043 |
| isomers_c7h8n2o2 | 0.947 ± 0.000 | 0.955 ± 0.008 | **0.969 ± 0.003** | 0.878 ± 0.026 | 0.852 ± 0.036 |
| isomers_c9h10n2o2pf2cl | **0.949 ± 0.025** | 0.850 ± 0.005 | 0.897 ± 0.007 | 0.865 ± 0.012 | 0.642 ± 0.054 |
| jnk3 | 0.793 ± 0.007 | **0.904 ± 0.004** | 0.764 ± 0.069 | 0.702 ± 0.123 | 0.783 ± 0.023 |
| median1 | **0.398 ± 0.000** | 0.340 ± 0.007 | 0.379 ± 0.010 | 0.257 ± 0.009 | 0.356 ± 0.009 |
| median2 | **0.357 ± 0.001** | 0.323 ± 0.005 | 0.294 ± 0.007 | 0.301 ± 0.021 | 0.276 ± 0.008 |
| mestranol_similarity | **0.939 ± 0.000** | 0.671 ± 0.021 | 0.708 ± 0.057 | 0.591 ± 0.053 | 0.618 ± 0.048 |
| osimertinib_mpo | **0.878 ± 0.007** | 0.866 ± 0.009 | 0.860 ± 0.008 | 0.844 ± 0.015 | 0.837 ± 0.009 |
| perindopril_mpo | **0.682 ± 0.013** | 0.681 ± 0.017 | 0.595 ± 0.014 | 0.547 ± 0.022 | 0.537 ± 0.016 |
| qed | **0.943 ± 0.000** | 0.939 ± 0.001 | 0.942 ± 0.000 | 0.941 ± 0.001 | 0.941 ± 0.000 |
| ranolazine_mpo | 0.716 ± 0.002 | **0.820 ± 0.016** | 0.819 ± 0.018 | 0.804 ± 0.011 | 0.760 ± 0.009 |
| scaffold_hop | **0.628 ± 0.018** | 0.576 ± 0.014 | 0.615 ± 0.100 | 0.527 ± 0.025 | 0.560 ± 0.019 |
| sitagliptin_mpo | **0.642 ± 0.000** | 0.601 ± 0.011 | 0.634 ± 0.039 | 0.582 ± 0.040 | 0.021 ± 0.003 |
| thiothixene_rediscovery | **0.642 ± 0.001** | 0.584 ± 0.009 | 0.583 ± 0.034 | 0.519 ± 0.041 | 0.534 ± 0.013 |
| troglitazone_rediscovery | **0.527 ± 0.008** | 0.448 ± 0.017 | 0.511 ± 0.054 | 0.427 ± 0.031 | 0.441 ± 0.032 |
| valsartan_smarts | **0.979 ± 0.003** | 0.627 ± 0.058 | 0.135 ± 0.271 | 0.000 ± 0.000 | 0.178 ± 0.358 |
| zaleplon_mpo | **0.591 ± 0.008** | 0.486 ± 0.004 | 0.552 ± 0.033 | 0.519 ± 0.029 | 0.358 ± 0.062 |
| Sum | **18.007** | 16.928 | 16.213 | 14.708 | 14.196 |

| Oracle | Graph-GA | SELFIES-REINVENT | GP-BO | STONED | LSTM-HC |
|---|---|---|---|---|---|
| albuterol_similarity | 0.838 ± 0.016 | 0.826 ± 0.030 | 0.898 ± 0.014 | 0.745 ± 0.076 | 0.719 ± 0.018 |
| amlodipine_mpo | 0.661 ± 0.020 | 0.607 ± 0.014 | 0.583 ± 0.044 | 0.608 ± 0.046 | 0.593 ± 0.016 |
| celecoxib_rediscovery | 0.630 ± 0.097 | 0.573 ± 0.043 | 0.723 ± 0.053 | 0.382 ± 0.041 | 0.539 ± 0.018 |
| deco_hop | 0.619 ± 0.004 | 0.631 ± 0.012 | 0.629 ± 0.018 | 0.611 ± 0.008 | 0.826 ± 0.017 |
| drd2 | 0.964 ± 0.012 | 0.943 ± 0.005 | 0.923 ± 0.017 | 0.913 ± 0.020 | 0.919 ± 0.015 |
| fexofenadine_mpo | 0.760 ± 0.011 | 0.741 ± 0.002 | 0.722 ± 0.005 | 0.797 ± 0.016 | 0.725 ± 0.003 |
| gsk3b | 0.788 ± 0.070 | 0.780 ± 0.037 | 0.851 ± 0.041 | 0.668 ± 0.049 | 0.839 ± 0.015 |
| isomers_c7h8n2o2 | 0.862 ± 0.065 | 0.849 ± 0.034 | 0.680 ± 0.117 | 0.899 ± 0.011 | 0.485 ± 0.045 |
| isomers_c9h10n2o2pf2cl | 0.719 ± 0.047 | 0.733 ± 0.029 | 0.469 ± 0.180 | 0.805 ± 0.031 | 0.342 ± 0.027 |
| jnk3 | 0.553 ± 0.136 | 0.631 ± 0.064 | 0.564 ± 0.155 | 0.523 ± 0.092 | 0.661 ± 0.039 |
| median1 | 0.294 ± 0.021 | 0.355 ± 0.011 | 0.301 ± 0.014 | 0.266 ± 0.016 | 0.255 ± 0.010 |
| median2 | 0.273 ± 0.009 | 0.255 ± 0.005 | 0.297 ± 0.009 | 0.245 ± 0.032 | 0.248 ± 0.008 |
| mestranol_similarity | 0.579 ± 0.022 | 0.620 ± 0.029 | 0.627 ± 0.089 | 0.609 ± 0.101 | 0.526 ± 0.032 |
| osimertinib_mpo | 0.831 ± 0.005 | 0.820 ± 0.003 | 0.787 ± 0.006 | 0.822 ± 0.012 | 0.796 ± 0.002 |
| perindopril_mpo | 0.538 ± 0.009 | 0.517 ± 0.021 | 0.493 ± 0.011 | 0.488 ± 0.011 | 0.489 ± 0.007 |
| qed | 0.940 ± 0.000 | 0.940 ± 0.000 | 0.937 ± 0.000 | 0.941 ± 0.000 | 0.939 ± 0.000 |
| ranolazine_mpo | 0.728 ± 0.012 | 0.748 ± 0.018 | 0.735 ± 0.013 | 0.765 ± 0.029 | 0.714 ± 0.008 |
| scaffold_hop | 0.517 ± 0.007 | 0.525 ± 0.013 | 0.548 ± 0.019 | 0.521 ± 0.034 | 0.533 ± 0.012 |
| sitagliptin_mpo | 0.433 ± 0.075 | 0.194 ± 0.121 | 0.186 ± 0.055 | 0.393 ± 0.083 | 0.066 ± 0.019 |
| thiothixene_rediscovery | 0.479 ± 0.025 | 0.495 ± 0.040 | 0.559 ± 0.027 | 0.367 ± 0.027 | 0.438 ± 0.008 |
| troglitazone_rediscovery | 0.390 ± 0.016 | 0.348 ± 0.012 | 0.410 ± 0.015 | 0.320 ± 0.018 | 0.354 ± 0.016 |
| valsartan_smarts | 0.000 ± 0.000 | 0.000 ± 0.000 | 0.000 ± 0.000 | 0.000 ± 0.000 | 0.000 ± 0.000 |
| zaleplon_mpo | 0.346 ± 0.032 | 0.333 ± 0.026 | 0.221 ± 0.072 | 0.325 ± 0.027 | 0.206 ± 0.006 |
| Sum | 13.751 | 13.471 | 13.156 | 13.024 | 12.223 |

## 1.3 NSGA-II SELECTOR

The NSGA-II (Deb et al., 2002) selector operates through three synergistic components to navigate multi-objective optimization landscapes:

- Fast non-dominated sorting (Algorithm 1) establishes hierarchical Pareto frontiers by iteratively categorizing solutions based on dominance relationships. Each solution's domination

count—the number of solutions superior in all objectives—and dominated set—solutions it outperforms—determine frontier membership. Solutions with zero domination counts form the first Pareto front, after which the algorithm recursively updates domination counts to construct subsequent fronts. This stratification prioritizes solutions closer to the true Pareto front while maintaining hierarchical selection pressure.

- Crowding distance computation (Algorithm 2) preserves diversity within each front by quantifying solution density in objective space. After sorting solutions along each objective axis, the metric calculates normalized distances between adjacent neighbors, assigning infinite values to boundary solutions. This ensures retention of extreme performers while promoting exploration of sparsely populated regions. Solutions with larger crowding distances receive priority during selection, effectively balancing convergence and diversity.

- The evolutionary workflow integrates these mechanisms through elitist population management. Parent and offspring populations merge before non-dominated sorting categorizes all solutions into Pareto fronts. Selection progresses front-by-front, starting with the highest-ranked solutions. When a front exceeds remaining population slots, crowding distance resolves the overflow by retaining the most spatially dispersed solutions. This dual strategy simultaneously advances the population toward optimality and prevents clustering in local optima, particularly crucial for molecular optimization where competing objectives like bioactivity and synthesizability require careful trade-offs.

---

**Algorithm 1** Fast Nondominated Sorting Approach of NSGA-II

**Input**: Population $P$
**Output**: Ranking nondominated solution set $F$

1: for each $q \in P$
2: $\quad S_p = \varnothing$
3: $\quad n_p = 0$
4: $\quad$ for each $q \in P$
5: $\quad\quad$ if $(p < q)$ then $\qquad\qquad$ If $p$ dominates $q$
6: $\quad\quad\quad S_p = S_p \cup \{q\}$ $\qquad\qquad$ Add $q$ to the set of solutions
$\qquad\qquad\qquad\qquad\qquad\qquad\qquad\qquad$ dominated by $p$
7: $\quad\quad$ else if $(q < p)$ then
8: $\quad\quad\quad n_p = n_p + 1$
9: $\quad$ if $n_p = 0$ then $\qquad\qquad\qquad p$ belong to the first front
10: $\quad\quad p_{\text{rank}} = 1$
11: $\quad\quad F_1 = F_1 \cup \{p\}$
12: $i = 1$ $\qquad\qquad\qquad\qquad\qquad$ Initialize the front counter
13: while $F_i \neq \emptyset$
14: $\quad Q = \varnothing$ $\qquad\qquad$ Used to store the member of the next front
15: $\quad$ for each $p \in F_i$
16: $\quad\quad$ for each $q \in S_p$
17: $\quad\quad\quad n_q = n_q - 1$
18: $\quad\quad\quad$ if $n_q = 0$ then $\qquad\quad q$ belongs to the next front
19: $\quad\quad\quad\quad q_{\text{rank}} = i + 1$
20: $\quad\quad\quad\quad Q = Q \cup \{q\}$
21: $\quad i = i + 1$
22: $\quad F_i = Q$

---

## 1.4 DETAILED PMO AUC SCORE RESULTS

This section provides the complete numerical data[1] that supports the analysis presented in the main text. Table 1 lists the detailed statistical results for the top-10 PMO AUC scores. The values are reported as the mean $\pm$ standard deviation over three independent runs. To facilitate comparison, the best-performing score for each case is highlighted in bold.

---

[1]The results are taken from (Lee et al., 2024).

**Algorithm 2** Crowding-distance Calculation of NSGA-II

**Input**: Population $P$
**Output**: Ranking nondominated solution set $F$

1: $l = |P|$    number of solutions in $P$
2: for each $i \in \{1, 2, \ldots, l\}$
3:      set $C_i = 0$    initial distance
4: for each objective $f_m$
5:        $P = \text{sort}(P, f_m)$    sort using each objective value
6:        $C_1 = C_l = \infty$    so that boundary points are always selected
7:        for $i = 2$ to $(l-1)$    for all other points
8:           $C_i = C_i + \frac{f_m(P_{i+1}) - f_m(P_{i-1})}{f_m^{\max} - f_m^{\min}}$    $f_m^{\max}$ and $f_m^{\min}$ denote the
                                     maximum and minimum
                                     values of the $f_m$

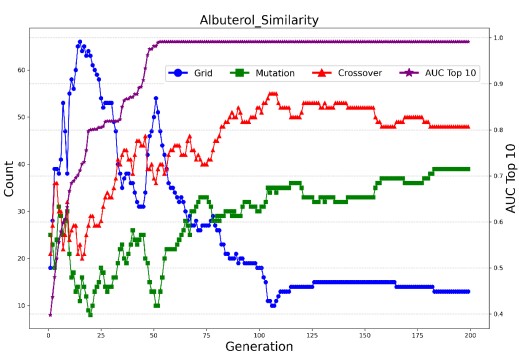

Figure 1: Operator Contribution and AUC Top 10 Performance on the Albuterol Similarity Task Across Generations. The x-axis represents the generation number, ranging from 0 to 200. The y-axis (left) represents the count of the different operators applied to the population: grid-based fragment-masked crossover (Grid), RL-based crossover (Crossover), RL-based mutation (Mutation). The y-axis (right) measures the Top-10 AUC score.

## 1.5 OPERATOR CONTRIBUTION ANALYSIS

The Figure 1 illustrates the results of the Albuterol Similarity task, showing the evolution of the population of molecules over 200 generations. The selection of offspring after each iteration is based on the NSGA-II selector, with various genetic operators contributing differently to the population. The plot tracks the frequency of each genetic operator used (Grid-based fragment-masked crossover, RL-based Mutation, and RL-based Crossover) as well as the corresponding AUC Top 10 score for the top candidates at each generation.

In the first few generations, the grid-based operator dominates (blue line), possibly helping to diversify the population. As the algorithm progresses, the mutation operator (green line) takes a more prominent role, indicating that introducing genetic variation is essential for the ongoing search for optimal solutions. During the mid-to-late stages, mutation and crossover operators continue to dominate, with the frequency of the grid-based operator tapering off. This shift suggests that as the population becomes more refined, the mutation and crossover operators are more effective in fine-tuning the candidates, improving the overall fitness of the population. The AUC Top 10 score steadily improves throughout the generations, reflecting the effectiveness of the applied operators in refining the population's performance. This suggests that the combination of mutation, crossover, and initial diversity from the grid operator facilitates the evolution of high-quality candidates over time.

## 1.6 TIME COMPLEXITY ANALYSIS

In each generation, the computational complexity of RL-GFM can be decomposed into three principal components. The foremost contributor is the NSGA-II backbone—comprising fast non-dominated sorting and subsequent crowding-distance assignment—whose cost scales as $O(MN^2)$, with $N$ denoting population size and $M$ the number of objectives. The second component is the grid-based ideal-point pairing module, which projects each of the $N$ individuals into a $5^M$-cell hyper-grid to identify both local and global ideal points, incurring $O(NM)$ complexity. Finally, the RL-guided variation operators introduce only constant overhead per individual: crossover policies are trained over a fixed candidate pool of size $N$, while mutation evaluates candidate solutions against a static library of $R = 89$ SMART reaction rules. Moreover, the RL component serves solely to accelerate the evolutionary algorithm's search process and does not engage in modeling the molecular distribution space; consequently, unlike many deep-learning–based methods, it does not require large-scale pretraining datasets (e.g., ZINC250k, comprising 250 000 molecules), thereby achieving substantially higher efficiency. Consequently, the dominant per-generation cost remains $O(MN^2)$, and over $G$ generations the total computational cost is $O(GMN^2)$, thus retaining the asymptotic efficiency of vanilla NSGA-II while embedding policy-driven operators at negligible extra expense.