# OpenReview forum: "Grid-Based Evolutionary Algorithm for Multi-Objective Molecule Generation Enhanced by Reinforcement Learning"
_ICLR.cc/2026/Conference — ICLR 2026 Conference Withdrawn Submission_

### Official Review · Reviewer_3RZY · 2025-10-19

**Soundness:** 2
**Presentation:** 3
**Contribution:** 2
**Rating:** 4
**Confidence:** 4

**Summary:**

This paper proposes **RL-GFM**, a reinforcement learning–driven grid-based evolutionary algorithm for multi-objective molecular generation. The method integrates two RL agents that guide crossover and mutation operators within a grid-based fragment-masked crossover framework, aiming to balance scaffold preservation with side-chain diversity while improving exploration efficiency. The optimization loop is built on NSGA-II, enabling multi-objective trade-offs across property optimization, synthesizability, and docking scores. Experiments are conducted on the 23 PMO benchmark tasks and several protein docking targets (PARP1, FA7, 5HT1B, BRAF, JAK2), showing that RL-GFM achieves state-of-the-art performance in both property optimization and docking, with higher novelty and diversity than baselines. An ablation study indicates that both the grid-based mechanism and the RL-guided operators contribute to the observed improvements. The authors position RL-GFM as a more interpretable, efficient, and robust alternative to prior fragment-based or black-box generative methods.

**Strengths:**

1. **Essential problem**: The paper tackles an important and timely challenge in molecular generation: balancing quality, diversity, and synthesizability under multi-objective constraints. By focusing on interpretable fragment-level operators and efficient oracle usage, it addresses practical limitations of existing fragment-based and black-box generative approaches.
2. **Fluent writing and logical flow**: The manuscript is generally well-written, with a clear logical progression from motivation to methodology, experimental setup, results, and conclusions. Figures and tables are well-structured and support the narrative effectively, making the contributions easy to follow.
3. **Abundant experiments and comparisons**: The experimental evaluation is extensive, covering both the PMO benchmark (23 tasks) and five protein docking targets. The paper compares against a wide range of strong baselines, and also includes ablation studies to validate the contributions of individual components.

**Weaknesses:**

### **Major**

1. **Method novelty**: The idea of **RL-based crossover and mutation** has already been explored in the Reinforced Genetic Algorithm (RGA) [1] for molecular generation, yet this prior work is not cited or discussed.
2. **Limitations of the grid-based method**: While the proposed **grid-based fragment-masked crossover** is novel, it has several drawbacks:
   - It intrinsically assumes a multi-objective setting. For single-objective problems, an artificial second objective (e.g., SA in the PMO benchmark) must be introduced, which may be unreasonable. For example, in PMO rediscovery tasks, the main goal is to find molecules similar to a target. If the target has poor SA, adding SA as an explicit objective may actively hinder optimization. In general, introducing arbitrary auxiliary objectives is not a principled solution.
   - The assumption of “intra-grid convergence” may not hold in practice, especially when optimal points in chemical space are sparse [2]. In many objectives such as docking scores, crossover offspring often perform worse than both parents. Thus, the theoretical premise behind the grid strategy is questionable for real-world property landscapes.
3. **Interpretability claim**: The paper repeatedly emphasizes interpretability, but it is unclear how the generated structures can be interpreted beyond standard fragment recombination. No concrete explanation or case study is provided to support this claim.
4. **PMO experimental setup**:
   - Several PMO objectives inherently conflict with diversity, such as rediscovery and similarity tasks, making the evaluation of diversity unreasonable in those settings.
   - Ten PMO objectives are themselves multi-property objectives (MPOs) originating from GuacaMol [3]. Since this paper emphasizes multi-objective molecular generation, it is unclear why those built-in multiple properties were not directly used as objectives, and instead a new SA score was introduced.
5. **Docking experimental setup**:
   - Comparisons are reported based on the top-5% molecules, but this subset may vary greatly in size across methods. A fixed set size (e.g., top 100) would make the comparison more consistent and fair.
   - No diversity metrics are reported for docking tasks, leaving it unclear whether RL-GFM achieves good diversity in addition to docking performance.

### **Minor**

1. No analysis of runtime or computational efficiency is provided.
2. Since no code or data is released, a reproducibility statement is recommended.
3. In Table 3, the standard deviations of RL-GFM results should be reported with the same precision (3 decimal places) as the baselines.

[1] Tianfan Fu, et al. "Reinforced Genetic Algorithm for Structure-based Drug Design." NeurIPS, 2022.

[2] Austin Tripp, et al. "An evaluation framework for the objective functions of de novo drug design benchmarks." ICLR workshop, 2022.

[3] Nathan Brown, et al. "GuacaMol: benchmarking models for de novo molecular design." JCIM, 2019.

**Questions:**

1. Could the authors clarify whether their grid-based crossover achieves unique benefits beyond efficiency compared to exhaustive parent pairing? In other words, are the gains due to pruning unpromising combinations, or does the grid introduce qualitatively new recombinations that would not appear in a full GA?
2. The initial population of molecules significantly determines the quality of the whole generation process, but this is not studied. For example, if the initial population doesn't contain a certain substructure, it will not be generated all along the fragment-based process. How to overcome this? In addition, the standard deviations in Table 3 are very low. Are they initialized with randomly selected different molecular sets?
3. For the PMO tasks, why you report the top-10 score AUC and the top-100 diversity, instead of a consistent set size? In addition, why the RL-GFM is evaluated under a budget of 3,000 oracle calls, instead of 10,000?

---

### Official Review · Reviewer_ZnvX · 2025-10-19

**Soundness:** 3
**Presentation:** 3
**Contribution:** 2
**Rating:** 6
**Confidence:** 2

**Summary:**

RL-GFM, a reinforcement learning and multi-objective evolutionary algorithm framework for molecular generation and optimization is proposed in this paper. The method integrates a grid-based fragment-masked crossover mechanism, RL-driven crossover and mutation policies. RL-GFM aims to balance multi-objectives during chemical space exploration. The authors extensively evaluate RL-GFM on PMO benchmark tasks and molecular docking tasks, and results demonstrate that RL-GFM achieves superior performance in most tasks.

**Strengths:**

1. Extensive experiments provide a robust validation of the method. RL-GFM outperforms most baselines in similarity-based and multi-property optimization tasks.

2. The manuscript is well-written and clearly structured.

**Weaknesses:**

1. Although the manuscript achieves strong results, it feels technically oriented, integrating Grid-based methods, RL, and multi-objective optimization, but lacks a clearly highlighted novel technical contribution.

2. Please refer to the Questions part.

**Questions:**

1. Based on the results in Table 3, RL-GFM shows a clear advantage in the mestranol_similarity and valsartan_smarts tasks, while in other tasks its advantage is relatively limited, and in some cases it even performs worse than other algorithms. Can you explain why this phenomenon occurs and why the algorithm performs better specifically in these two tasks?

2. As stated in Line 239, "The crossover policy determines parent pair molecules, which are then split into fragments and randomly recombined to form new molecules". Could you compare how this randomness affects the algorithm? Is its performance robust across multiple runs, and what is the variance?

3. Ablation experiments on the hyperparameters need to be conducted to assess the robustness of the algorithm, for example, for $\epsilon$, $k$, etc.

4. Compared to other algorithms, what are the advantages of this algorithm in terms of efficiency and real-time computational cost?

---

### Official Review · Reviewer_iZFB · 2025-10-21

**Soundness:** 2
**Presentation:** 3
**Contribution:** 3
**Rating:** 4
**Confidence:** 4

**Summary:**

The paper introduces **RL-GFM**, a reinforcement learning–driven grid-based evolutionary algorithm for molecular generation. The framework combines an NSGA-II multi-objective optimizer with two RL-guided operators for crossover and mutation, alongside a grid-based fragment-masked crossover strategy designed to preserve scaffolds while diversifying side chains. The approach is motivated by the limitations of fragment-based drug discovery and black-box generative models, aiming for more interpretable and efficient exploration of chemical space.

The authors evaluate RL-GFM on 23 PMO benchmark tasks and on docking experiments against five protein targets (PARP1, FA7, 5HT1B, BRAF, JAK2). Results show that the method achieves the highest cumulative optimization score across PMO tasks, with notable improvements in similarity and multi-property optimization challenges, and demonstrates superior balance among diversity, novelty, and synthesizability. In docking experiments, RL-GFM delivers the best mean binding affinities across all targets and a higher ratio of novel hits compared to state-of-the-art baselines. Ablation studies further confirm that both the grid-based method and RL-guided operators contribute substantially to performance. Overall, RL-GFM demonstrates strong empirical results and highlights the potential of combining RL and evolutionary search for multi-objective drug design.

**Strengths:**

The paper's main strengths lie in its well-motivated methodological contributions and strong empirical validation. First, it introduces a novel integration of RL-guided crossover and mutation operators within an NSGA-II optimization framework, which enables more adaptive and targeted exploration of chemical space compared to purely stochastic evolutionary search. Second, the grid-based fragment-masked crossover is a creative mechanism that preserves core scaffolds while systematically diversifying side chains, effectively balancing convergence and diversity in multi-objective optimization. Third, the extensive experiments on both PMO benchmarks and docking tasks highlight clear improvements over state-of-the-art baselines, demonstrating that the proposed approach not only achieves higher optimization scores but also discovers more novel and synthesizable compounds. These contributions collectively show that RL-GFM is a promising framework for advancing multi-objective molecular generation.

**Weaknesses:**

- **PMO benchmark setup**: Some PMO objectives are inherently contradictory with diversity (e.g., rediscovery). Reporting diversity alongside such objectives can be misleading, since the optimal solutions by definition are structurally constrained.
- **Interpretability claim**: The contribution of interpretability is not well supported. Although mentioned in the Introduction, Related Work, and Conclusion, the Methods and Experiments sections do not provide any explanations, metrics, or case studies to illustrate how the approach enhances interpretability in practice.
- **Relation to prior work (RGA)**: The combination of reinforcement learning with genetic algorithms for molecular design has already been introduced in RGA (*Reinforced Genetic Algorithm for Structure-based Drug Design*, NeurIPS 2022). This work is not cited or discussed, which risks overclaiming novelty.
- **Typos and formatting issues**:
  - Line 98: "modifing" should be "modifying"
  - Figure 1: "offspring moleculse" should be "offspring molecules"
  - Table 3: The standard deviations of RL-GFM results are not consistently reported to three decimal places.

**Questions:**

- Could the authors provide evaluation curves showing performance over iterations of the RL process? This would help illustrate convergence behavior and stability during training.
- The grid-based strategy partitions the objective space by properties, yet the underlying search operates on molecular structures. Given that structure–property relationships can be highly non-convex and complex, how do the authors justify the effectiveness of applying a grid-based approach, which is more common in convex optimization contexts?
- Why is the Synthetic Accessibility (SA) score incorporated as an additional oracle in the PMO experiments? How does SA relate to the original PMO objectives, and could this addition potentially bias the optimization outcomes?

---

### Official Review · Reviewer_vnzS · 2025-10-28

**Soundness:** 2
**Presentation:** 2
**Contribution:** 1
**Rating:** 2
**Confidence:** 4

**Summary:**

Through this paper, the authors propose Reinforcement Learning-Driven Grid-based Fragment-Masked Multi-objective Evolutionary Algorithm (RL-GFM). The authors claim that RL-GFM can overcome the limitations of existing FBDD methods, the need to construct and maintain static fragment libraries.

**Strengths:**

- The concept figure aids in understanding the work.
- The writing is easy to follow.

**Weaknesses:**

My concerns are as follows:
- The main weakness of this paper is its weak novelty. As the authors mentioned in lines 137~140, the proposed RL-GFM framework comprises three components: (1) a grid-based fragment-masked crossover operator, (2) RL-driven crossover and mutation operators, and (3) a Pareto-optimal solution selector.
	- However, in the grid-based fragment-masked crossover (Section 4.1) is a heuristic and cannot be considered a significant contribution from an ML perspective. Moreover, the idea is very similar to the fragment remasking of GenMol [1], but there is no detailed comparison with this work.
	- The idea of RL-based genetic operations (Section 4.2) is very similar to those of [2] and [3].
	- The idea of the Non-dominated Sorting Genetic Algorithm II (NSGA-II) algorithm to perform multi-objective molecular optimization is from [4] and is not an invention of this work. Overall, I am not convinced that this work provides a novel approach compared to previous methods in the domain.
- The authors claim that the limitations of existing FBDD methods are (1) reliance of static fragment libraries, (2) inefficient construction and maintenance of fragment libraries, and (3) lack of interpretability. However, several existing methods have already overcome these limitations. For example, GEAM [5], one of the baselines in this paper, addresses the limited exploration problem by introducing a dynamic fragment library, enables very fast fragment library generation, and provides an interpretable fragment library.
- In the PMO experiment (Section 4.1, Figure 3), SOTA molecular optimization baseline, GenMol [1], is missing.

---

**References:**

[1] Lee et al., GenMol: A Drug Discovery Generalist with Discrete Diffusion, ICML 2025.

[2] Fu et al., Reinforced Genetic Algorithm for Structure-based Drug Design, NeurIPS, 2022.

[3] Ahn et al., Guiding Deep Molecular Optimization with Genetic Exploration, NeurIPS, 2020.

[4] Verhellen, Graph-based molecular pareto optimisation. Chemical Science, 2022.

[5] Lee et al., Drug discovery with dynamic goalaware fragments, ICML, 2024.

**Questions:**

Please see the *Weaknesses* section for my main concerns.
- Why are the results for f-RAG, the best-performing baseline in Figure 3 and Table 3, missing from Table 2 and Table 4?

---

### Note · Authors · 2025-11-21

I have read and agree with the venue's withdrawal policy on behalf of myself and my co-authors.